# Protocol for process evaluation of ARTEMIS cluster randomised controlled trial: an intervention for management of depression and suicide among adolescents living in slums in India

Ankita Mukherjee [1] ,[1] Sandhya Kanaka Yatirajula [1] ,[1] Sudha Kallakuri,[2] Srilatha Paslawar [1] ,[2] Heidi Lempp,[3] Usha Raman,[4] Ashok Kumar,[5] Beverley M Essue,[6] Rajesh Sagar,[7] Renu Singh,[8] David Peiris [1] [9,10] Robyn Norton,[9,11] Graham Thornicroft,[12] Pallab Kumar Maulik [1] [1,10]

For numbered affiliations see end of article.

**Correspondence to**
Dr Pallab Kumar Maulik;
pmaulik@georgeinstitute.org.in

## ABSTRACT

**Introduction** There are around 250 million adolescents (10–19 years) in India. The prevalence of mental health-related morbidity among adolescents in India is approximately 7.3%. Vulnerable subpopulations among adolescents such as those living in slum communities are particularly at risk due to poor living conditions, financial difficulty and limited access to support services. Adolescents' Resilience and Treatment nEeds for Mental Health in Indian Slums (ARTEMIS) is a cluster randomised controlled trial of an intervention that intends to improve the mental health of adolescents living in slum communities in India. The aim of this paper is to describe the process evaluation protocol for ARTEMIS trial. The process evaluation will help to explain the intervention outcomes and understand how and why the intervention worked or did not work. It will identify contextual factors, intervention barriers and facilitators and the adaptations required for optimising implementation.

**Methods** Case study method will be used and the data will include a mix of quantitative metrics and qualitative data. The UK Medical Research Council's guidance on evaluating complex interventions, the Reach, Efficacy, Adoption, Implementation and Maintenance Framework and the Affordability, Practicability, Effectiveness and cost-effectiveness, Acceptability, Safety/Side Effects and, Equity criteria will be used to develop a conceptual framework and a priori codes for qualitative data analysis. Quantitative data will be analysed using descriptive statistics. Implementation fidelity will also be measured.

**Discussion** The process evaluation will provide an understanding of outcomes and causal mechanisms that influenced any change in trial outcomes.

**Ethics and dissemination** Ethics Committee of the George Institute for Global Health India (project number 17/2020) and the Research Governance and Integrity Team, Imperial College, London (ICREC reference number: 22IC7718) have provided ethics approval. The Health Ministry's Screening Committee has approved to the study (ID 2020-9770).

## STRENGTHS AND LIMITATIONS OF THIS STUDY

⇒ This study will use existing implementation science theories and frameworks and qualitative and quantitative data will be triangulated to arrive at a comprehensive understanding of the intervention.
⇒ This study will be guided by the Medical Research Council framework for developing and evaluating complex interventions and the case study method will also be used.
⇒ Cases will be purposively selected based on maximum variation approach. Both intervention and control sites will be selected as cases to the enable comparison, to understand contextual factors and to avoid the Hawthorne effect.

**Trial registration number** CTRI/2022/02/040307.

## INTRODUCTION

India has about 250 million adolescents aged 10–19 years, comprising nearly one-fifth of India's population.[1] The burden of mental health problems in adolescents is a growing concern globally.[2] In India, the prevalence of psychiatric disorders among adolescents is estimated to be about 7.3%.[3] Self-harm and depressive disorders are the leading cause of death and disability in the age group 15–19 years.[4]

Mental health interventions for adolescents are important as 50% of adult mental disorders have an onset before the age of 14 years.[5] Marginalised populations, such as adolescents living in slum communities, are particularly vulnerable because of additional

stressors related to poor living conditions, financial stress and poor access to support services.[6 7]

There has been limited research to test community-based mental health interventions for adolescents living in slums in India. A scoping review of mental health interventions among adolescents in India found that of the 11 interventions included in the review, 9 were school based, 1 community based and 1 was digital.[8] Most of the school-based programmes used a life skills curriculum that resulted in improvements in depressive symptoms and overall mental well-being. The review recommended the need for more interventions for early and out-of-school adolescents to ensure that the most vulnerable adolescents were not missed out. An intervention to build mental health and resilience delivered by community-based peers among highly disadvantaged young women living in urban slums in Dehradun, a city in north India, showed sustained improvements in anxiety and depression and attitudes to gender equality among study participants.[9] A randomised controlled trial of a 5-month resilience-based programme among rural adolescent girls through government schools in the state of Bihar, India, delivered by local women showed that girls receiving the intervention (vs controls) had better emotional resilience, self-efficacy, social-emotional assets, psychological well-being and social well-being.[10] Another school-based pilot study of of POD (Problems, Options, Do it), titled Adventures (a smartphone-based blended problem-solving game-based intervention for adolescents with or at risk of anxiety, depression and conduct difficulties) was helpful in managing their problems and stress and improving the mental health of 13–19 years enrolled in secondary schools in the Indian state of Goa.[11] The Adolescents' Resilience and Treatment nEeds for Mental Health in Indian Slums (ARTEMIS) cluster randomised control trial (cRCT) is testing a community-based intervention to improve the mental health outcomes for adolescents living in urban slum clusters in India.

This paper presents the protocol for process evaluation of the ARTEMIS cRCT. ARTEMIS is a community-based cluster randomised control trial that aims to reduce depression and the risk of suicide among adolescents living in slums. The intervention will use a mental health stigma reduction campaign with adolescents and parents of the study cohort to improve attitudes towards mental health and improve help seeking. A technology-enabled strategy will be employed for screening, clinical diagnosis and management of mental health problems (depression, other significant emotional or medically unexplained complaints and suicide risk) among adolescents by primary care doctors and community-based non-physician health workers (NPHWs) (described in detail below).

Process evaluations provide critical inputs in understanding how interventions work in particular contexts and thus, support implementation planning beyond the trial setting.[12]

## Aims
The aims of the process evaluation are to:
1. Assess implementation fidelity and understand how the intervention was implemented.
2. Identify key contextual factors that impact intervention delivery and outcomes.
3. Understand perceptions of different stakeholders about effectiveness, acceptability and scalability of intervention components.
4. Identify key facilitators and barriers in implementation of the intervention.
5. Explain any adaptations to the intervention or intervention refinement during the study and their possible impact on the outcomes.

## METHODS
### Conceptual framework
The process evaluation will be guided by the Medical Research Council (MRC) framework for developing and evaluating complex interventions.[13 14] The framework highlights the importance of examining the implementation, the mechanism of impact and their interaction with contextual factors to better understand how and why an intervention does or does not work. A recent update to the framework recognises that complex interventions have several phases including intervention development, feasibility assessment, implementation and evaluation, which may not always be sequential.[15] It recommends six core areas of inquiry at each phase before moving on to the next phase. They include (1) the intervention and its interaction with context; (2) the programme theory; (3) ways of engaging with diverse stakeholders; (4) ways of intervention refinement; (5) identifying key uncertainties and (6) economic considerations. The process evaluation will focus on the first five areas of inquiry, and the overall objectives have been framed to address these key areas.

### Study setting
The ARTEMIS cRCT will be implemented in 60 slum clusters across two cities New Delhi and Vijayawada in India. In each city, 30 slum clusters will be included. For this study, a slum cluster is defined as slums within wards or geographical areas identified as slums/resettlement colonies. A ward is a local authority area, typically used for electoral purposes. In certain cities of India, such as Mumbai and New Delhi, a ward is an administrative unit of the city region; a city area is divided into zones, which in turn contains numerous wards. New Delhi is a metropolis and one of the largest cities in India with a population of about 17 million and an estimated slum population of about 2 million.[16 17] Vijayawada is one of the largest cities in the state of Andhra Pradesh with an urban population of over 1.0 million[18 19] and an estimated slum population is estimated to be about 0.5 million.[17] The most widely spoken language in Delhi is Hindi while in Vijayawada it is Telugu.

## Patient and public involvement

Adolescent Expert Advisory Groups (AEAGs) have been formed at each site. This group was involved throughout the intervention development phase in providing inputs and suggestions. Their contributions were critical for the cocreation of the anti-stigma content for adolescents. During the formative phase, 34 meetings were held where the AEAG provided valuable feedback on ways to engage adolescents in the anti-stigma campaign.

## Study design

The study will use a mixed-method, multiple case study design. Each slum cluster will constitute a 'case' for the study. A total of six cases or clusters will be purposively selected taking a maximum variation approach. Slums will be purposively selected to represent different contexts, coverage and reach of intervention as well as the ease or difficulty of implementing the intervention.

## Intervention description

The ARTEMIS cRCT will test an intervention to address depression, increased risk of self-harm/suicide or other significant emotional or medically unexplained complaints among adolescents living in urban slums in two cities in India. ARTEMIS has two components. The first is a campaign that aims to reduce stigma related to mental health and improve attitudes and behaviours towards adolescents with depression or at increased risk of self-harm/suicide. The second is a technology-enabled mHealth platform with an integrated electronic decision support system (EDSS), to help primary care doctors and NPHWs to diagnose and treat adolescents at high risk of depression, self-harm or suicide.

Before randomisation and the start of the intervention, a team of trained field investigators will screen adolescents at high risk of depression or suicide using Patient Health Questionnaire-9 (PHQ-9), which is a standardised psychometric tool for screening depression and suicide risk in the community.[20 21] Adolescents who obtain a PHQ 9 score of ≥10 and/or a score of ≥2 in the suicide risk question on the PHQ-9 will be deemed as 'high risk'. Due to the time delay between screening and randomisation and potential natural remission, a second screening will be carried out for adolescents identified as 'high risk' before the baseline is administered. This process will help identify the final list of adolescents at 'high risk' in all the clusters. The following information will be collected from the study cohort by the trained field investigators: sociodemographic characteristics, history of mental illness, treatment history, comorbid conditions, stressful events experienced in the previous year, resilience, knowledge attitude and behaviours related to mental health and stigma associated with help seeking. A detailed protocol of the trial has been published.[22]

A theory of change model for the intervention was developed with key stakeholders during the formative phase of the study in 2021. The logic model for intervention has been provided (figure 1).

## Intervention arm

In the intervention arm, the two components (1) the anti-stigma campaign and (2) the technology-enabled mHealth-based EDSS for screening, diagnosis and management of depression and self-harm will be delivered. (figure 1).

### Anti-stigma campaign

The anti-stigma campaign will include several Information Education and Communication (IEC) materials containing key messages around addressing mental health stigma, myths related to mental health, highlighting major stressors and their impact on mental health of adolescents and the importance of help seeking for mental health problems. The IEC materials will include posters, pamphlets, brochures, videos of persons with lived experience, animation videos, audio dramas, games, magic shows, rallies and street plays. The materials will be cocreated with adolescents from the study sites. AEAGs will be created in both locations consisting of adolescents from the community, with the purpose of involving them in working with the implementation team to cocreate the anti-stigma campaign. Details of the process of forming AEAGs and their engagement in shaping the anti-stigma campaign have been published elsewhere.[23]

### Peer-educators

Based on findings from the formative phase of the study, peer groups were established comprising adolescents from the community in the age group 13–19 years, some of whom were from the study cohort. Their role will be to provide support to adolescents who approach them to discuss problems and to promote mental health awareness in the community. All the members of the peer groups will be provided basic training by the research team on mental health, safe practices while using social media, addiction and substance use and promoting mental wellbeing through engaging in pleasurable activities such as listening to music, sharing problems with trusted individuals, having friends, exercising and eating healthily.

### Technology-enabled EDSS

The EDSS platform has been developed to assist primary care doctors and NPHWs to screen, diagnose and manage depression, other significant emotional or medically unexplained complaints and suicide risk among adolescents. High-risk adolescents will be referred to the doctor electronically via the mHealth platform by the NPHWs. In each cluster, one doctor and several NPHWs will be trained. The NPHWs will be trained to use the EDSS to provide basic supportive advice to adolescents at high risk of depression or self-harm, refer them to the primary care doctors, and track treatment advice given by the doctors and will follow-up on treatment adherence. The NPHWs will make regular home visits to encourage adolescents at high risk and their families to seek help from the doctor. The NPHWs will also use the EDSS to clinically monitor the severity of high-risk individuals at 4 months and at

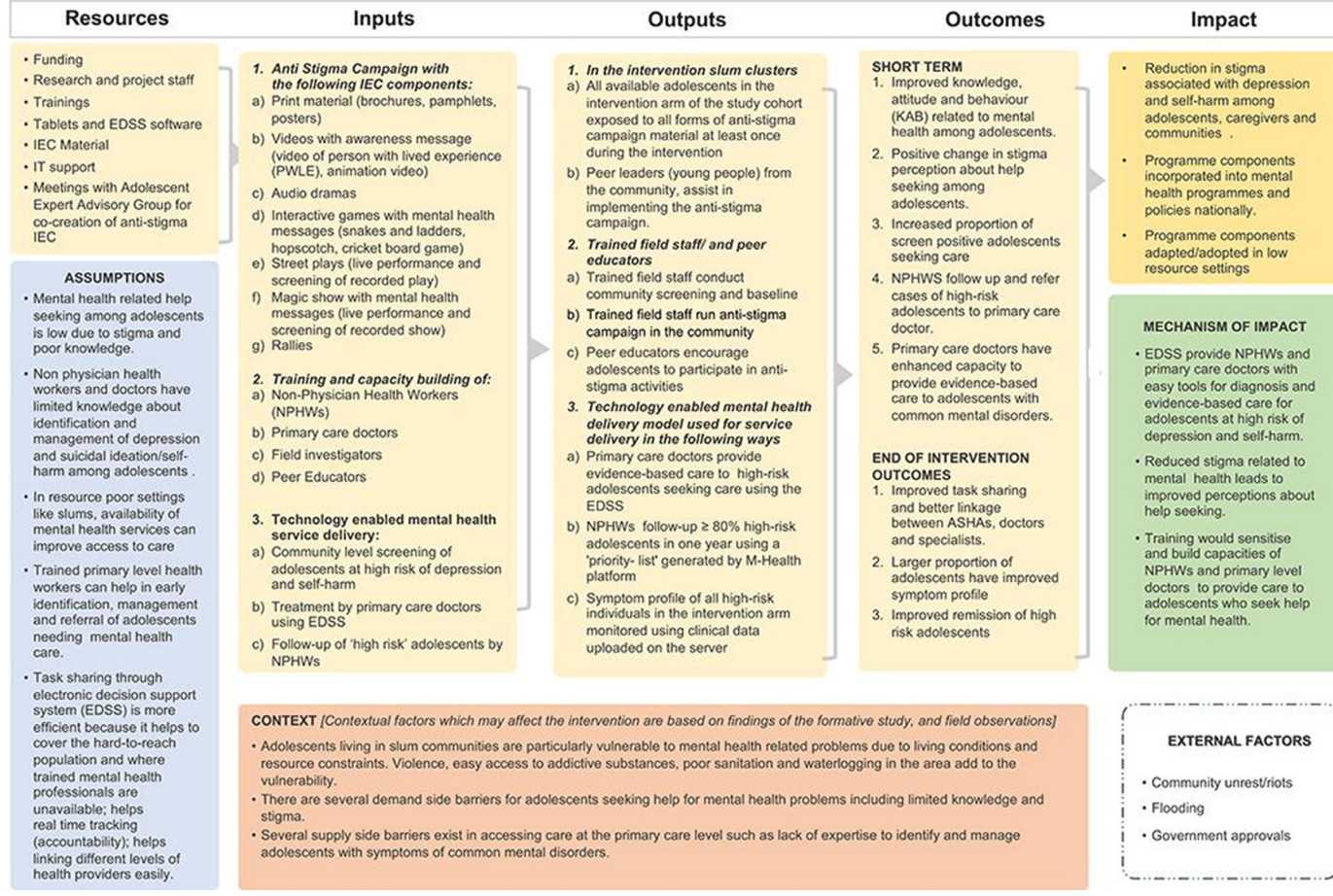

**Figure 1** Logic model for ARTEMIS. IEC, Information Education,Communication; IT, Information Technology; ASHAs, Accredited Social Health Activists.

10 months after the start of the intervention. High-risk individuals will be provided referral cards to take to the primary care doctors. The primary care doctors will be trained to use an EDSS to clinically diagnose and manage adolescents with depression, other significant emotional or medically unexplained complaints and self-harm/suicide ideation based on WHO's mhGAP (Mental Health Gap Action Programme)-Intervention Guide algorithm.[24] Adolescents needing specialist care will be referred to a psychiatrist in a government facility. The EDSS will also capture and store (on a cloud server) information regarding adolescents' visits and the type of care provided by the primary care doctor—counselling/pharmacotherapy/combination of both/referral/follow-up. All data will be encrypted and protected using user-defined passwords. The information will be visible to the NPHWs, the project supervisors and research team only. A colour-coded traffic light system integrated in the application will assist the NPHWs to identify high-risk adolescents who need to be prioritised for follow-up. The EDSS will also provide them with simple questions tailored to the priority level as indicated by the traffic-light coding system, that they can ask the adolescents during follow-up to ensure appropriate treatment adherence or follow-up with doctors or mental health specialists.

**Table 1** Qualitative data collection plan

| Intervention arm | | |
|---|---|---|
| Data sources | FGDs planned | Interviews planned |
| Non-physician health workers (NPHWs) | 4 | – |
| Adolescents from high-risk cohort | 8 | – |
| Adolescents from non-high-risk cohort | 4 | – |
| Parents of high-risk adolescents in the study cohort | | 24 |
| Members of Adolescent Expert Advisory Groups | 5 | – |
| Peer leaders | 4 | – |
| Field staff | 2 | – |
| Key implementation team members | 1 | |
| Primary care doctors | – | 4 |
| Control arm | | |
| NPHWs | 2 | – |
| Adolescents from study cohort | 4 | – |
| Total | 34 | 28 |

FGDs, focus group discussions.

**Table 2** Conceptual Framework for Process Evaluation

| Broad area of enquiry | Domains of inquiry | Key questions/Process Measures | Data source |
|---|---|---|---|
| Context | Differences in context | ► What are the differences in terms of geography, slum conditions, socio-cultural norms, economic between the sites and among the clusters?<br>► What are the contextual factors which affect programme delivery, stakeholder response and intervention uptake in different settings?<br>► Who were the key stakeholders who could play a role in improving intervention outcomes? | Secondary data;<br>Formative research data<br>FGDs with project staff<br>FGDs with key members of the implementation team |
| | Intervention adaptions based on context | ► What are some of the key contextual factors which influenced the overall implementation of the intervention?<br>► What were some of the context- specific adaptations that were made to address emerging challenges? | FGDs with project staff<br>FGDs with key members of the implementation team<br>project documentation completed during the project |
| | Contextual barriers and facilitators | ► What are key facilitators in different contexts that helped in intervention delivery and uptake?<br>► What are the barriers in various contexts for implementing the intervention components? | FGDs with project staff<br>FGDs with key members of the implementation team |
| Implementation | Implementation fidelity | Was the intervention delivered as it was planned? | Program records and documents;<br>observation and rating |
| | Intervention reach | ► What was the coverage of the different anti-stigma campaign methods, in terms of:<br>– Total persons reached (including age and gender-wise break-up)<br>– Slums and clusters covered.<br>– Number and proportion of high-risk cohort reached.<br>– Number and proportion of non-high-risk cohort reached.<br>– Key stakeholders reached.<br>► What was the reach of the mHealth services: Number and proportion of high-risk cohort from the intervention arm who sought care with the primary care doctor.<br>Number and proportion high risk-cohort from the control arm who sought care.<br>Number and proportion of high-risk cohort in the intervention arm provided counselling or follow-up services by NPHWs.<br>► Did the NPHWs face any challenge in reaching out adolescents and their families during the intervention? | Project records and documents<br>backend data<br>FGDs with project staff FGDs with key members of the implementation team<br>Interviews with NPHWs |
| | Intervention effectiveness | ► What was the perception of adolescents and key stakeholders about the utility effectiveness of the anti-stigma campaign?<br>► What are some of the key take home messages from anti stigma campaign for adolescents and community members?<br>► What was the perception of NPHWs about impact of anti-stigma campaign in their community?<br>► What is the association between exposure to anti stigma content with changes in KAB scores and care seeking?<br>► What is the perception of NPHWs about effectiveness of mHealth-based EDSS for management of depression and self-harm in adolescents?<br>► What is the perception of doctors about effectiveness of mHealth-based EDSS for community-based management of depression andself-harm in adolescents? | Community satisfaction survey done at the end of street play performance.<br>Outcome survey data;<br>backend data<br>FGDs with adolescents<br>FGDs with AEAG and peer leaders<br>Interviews with parents |

**Table 2** Continued

| Broad area of enquiry | Domains of inquiry | Key questions/Process Measures | Data source |
|---|---|---|---|
| | Intervention acceptability and adoption | ▶ What was the experiences of NPHWs in using EDSS for providing care (challenges, perceived benefits, potential for routine use of mHealth)?<br>▶ What was the experiences of doctors about using EDSS for providing care (challenges, perceived benefits, potential for routine use of mHealth)?<br>▶ What were some key features of use of EDSS by doctors:<br>– Average time taken for diagnosis and identification of treatment plan using mhGAP over time.<br>– Association between type of CMD and time taken for diagnosis and identification of treatment plan using mhGAP.<br>▶ What were the perceptions of high-risk adolescents and their parents about ease of getting treatment through mHealth? | Backend data FGDs with NPHWs Interviews with doctors Interview with high-risk adolescents and their parents |
| | Practicability | ▶ What were the challenges/barriers in delivering of the different strategies of anti-stigma campaign to adolescents?<br>▶ What were the challenges of the electronic decision support systems by NPHWs and doctors for diagnosis and management of depression and self-harm among adolescents?<br>▶ Doctor's experiences of interactions with adolescents<br>▶ Experiences of NPHWs with adolescents during follow up and motivating them and their families to visit the doctor initially and for follow up | FGDs with NPHWs Interviews with doctors FGDs with project staff |
| | Safety and side Effects | ▶ Were there any unanticipated negative consequences because of the antistigma campaign?<br>▶ Were there any safety concerns related to using mHealth based EDSS platform for management of depression and selfharm among adolescents? | FGDs with adolescents FGDs with NPHWs Interviews with doctors FGDs with project staff |
| Mechanism of impact | Variation in outcomes | What kind of cluster level variation is observed in in the outcomes? What works, for whom and in what context? | Outcome data Backend data; FGDs with NPHWs Interviews with doctors FGDs with project staff |
| | Unexpected outcomes | What are some unexpected outcomes and what factors can be attributed to them? | Outcome data back-end data; FGDs with NPHWs Interviews with doctors FGDs with project staff |
| | Theory of change (ToC)/ pathways to achieving outcomes | What were some of the key pathways to improved outcomes for high-risk adolescents?<br>What changes and adaptations were made to the ToC during the intervention?<br>What were some of the key assumptions regarding causal mechanisms that need to be modified? | Outcome data back-end data Project documents FGDs with project staff |

AEAG, Adolescent Expert Advisory Group; EDSS, electronic decision support system; FGDs, focus group discussions; KAB, Knowledge Attitude Behaviour; mhGAP, Mental Health Gap Action Programme; NPHWs, non-physician health workers.

### Control arm

The control arm will receive enhanced usual care that will include general psychoeducation through pamphlets to raise awareness of common mental disorders in adolescents among the wider community. Parents/guardians of adolescents identified as high risk will be advised to consult a primary care doctor or mental health professional. Those with high scores on PHQ-9 (a score of ≥15 on PHQ9) and/or at high risk of suicide (suicide score ≥2), indicating severe depression and/or high risk of suicide will be asked to seek care immediately and will be given a list of health facilities (with contact details) where such treatment is available. All NPHWs will be asked to follow up with the adolescents and their families more intensively and will recommend that they visit a doctor/psychiatrist immediately.

### Data collection

Quantitative data for the process evaluation will draw on metrics collected and sourced from the open-source

medical record system or OpenMRS; (OpenMRS.org). The OpenMRS is a standardised community-driven open-source software for storing and processing medical record information. Quantitative data will be collected throughout the intervention via the mHealth platform. This will include data on screening outcome, treatment seeking, follow-ups and treatment provided for the high-risk cohort. Data will also be used to understand usage patterns of the EDSS by primary care doctors and NPHWs. The mHealth platform will capture this data in tablets used by NPHWs and primary care doctors for screening, treatment and follow-up of the high-risk cohort. These data will be used to assess reach, effectiveness and service utilisation. Data on competency and fidelity measures will also be captured throughout the intervention.

Qualitative data collection will include key informant interviews (KIIs) and focus group discussions (FGDs) with various stakeholders including high-risk and non-high-risk cohorts in both arms, parents, primary care doctors, NPHWs, members of the AEAG and peer leaders. A total of 33 FGDs (each with 6–12 participants) and 28 interviews are planned across the 6 purposively selected slum clusters equally divided between both sites. This will include four intervention clusters (two each in Delhi and Vijayawada) and two control clusters (one each in Delhi Vijayawada) (table 1). The KIIs and FGDs with stakeholders will help capture information related to fidelity of the intervention and identify any gaps that could be addressed. Written informed consent will be sought from all participants selected for KIIs and FGDs.

### Data analysis

The conceptual framework for the study and how it integrates with the key parameters from the MRC frameworks is provided in table 2.

The case study methodology will also be used as it is recognised to be suitable to study complex interventions in different contexts and helps to capture 'the complexity of the case, the relationship between the intervention and the context and how the intervention worked (or did not)'.[25] Following Pfadenhauer et al, we define context as 'a set of characteristics and circumstances that consist of active and unique factors that surround the implementation. As such it is not a backdrop for implementation but interacts, influences, modifies and facilitates or constrains the intervention and its implementation'.[26] We will be selecting both intervention and control sites as cases to avoid the Hawthorne effect. Two slum cluster in the control arm will also be selected for the case study to enable comparison, to understand contextual factors and any changes to usual care in the absence of an intervention. The importance of studying changes in control arms has been recognised as being useful in understanding intervention impact.[27] A multiple case study design will be used where each case will be analysed at the case level, but we will also pull together all the cases from the intervention and control arms for a more in-depth analysis and comparison. Basic descriptive analysis will be conducted

with the quantitative data. Qualitative data will be transcribed by a professional vendor and the transcripts will be read by at least two members of the research team and coded. Coding will be carried out using a priori codes based on the conceptual framework (table 3) with the help of NVivo 12 software.

Any additional codes emerging during the analysis will be added to the coding framework. Members of the research team will read transcripts separately and come up with codes. The research team will then compare and combine their codes to evaluate their fit and usefulness and will examine the differences in code to see if any new insights can be generated. The code list will be finalised after a discussion between the researchers to establish agreement among coders. Qualitative and quantitative data will be triangulated to arrive at a comprehensive understanding of the intervention. Triangulation helps in confirming the findings from quantitative and qualitative data thereby increasing validity as well as leading to a better understanding of phenomena being studied.[28]

The quantitative data will be used understand implementation outcomes. Indicators to study implementation outcomes will be informed by the conceptual framework and relevant implementation science theories and frameworks (table 2). These include the RE-AIM[29] (Reach, Efficacy, Adoption, Implementation and Maintenance) framework and the APEASE (Affordability, Practicability, Effectiveness and cost-effectiveness, Acceptability, Safety/Side Effects and, Equity) criteria for evaluation of behaviour change interventions.[30–32] The APEASE criteria have some similarities with RE-AIM but includes two additional parameters, which are- practicability and safety. An intervention is practicable if 'it can be delivered as designed through the means intended to the target population'.[30] Data will be extracted from OpenMRS and descriptive statistics will be used to get totals, proportions and changes over time.

### Assessment of implementation fidelity

Implementation fidelity refers to the extent to which an intervention was implemented as intended. We will assess fidelity in three components, such as (a) delivery of anti-stigma campaign, (b) implementation of the mHealth component and (3) trainings. Indicators have been developed for all these components to measure the frequency of exposure and coverage. The quality of training provided to primary doctors on the mhGAP, NPHWs on the priority listing app and peer educators on promoting mental well-being and competency will be assessed through various tools including a post-training satisfaction survey, pre–post test self-assessment checklists as well as though rating by trainers and staff on competency checklists.

### Discussion

This paper describes the design of a mixed-method process evaluation for the ARTEMIS cRCT, which

**Table 3** Data Sources and Areas of Inquiry for Qualitative Data Collection

| Type of group/ individual | Some areas of inquiry |
|---|---|
| Non-physician health workers | Perceptions on effectiveness and appropriateness of training received. Experience of using mHealth platform for delivering the intervention including challenges or facilitators Perception about effectiveness anti-stigma campaign in improving help seeking. Facilitators and barriers in treatment seeking by high-risk cohort. Overall experience of participating in the trial. Perception about any personal benefits/harms due to their association with the project. Perception about what role they see for themselves in the community once the project comes to an end. Any spillover effects-experiences of helping or sharing information with other adolescents not in the study cohort. |
| Project field staff in each site | Barriers or facilitators in delivering anti-stigma campaign in the community. Barriers and facilitators in delivering mHealth component. Perceived factors which explain high/low treatment seeking in different clusters. Key lessons learnt and suggestions for implementation of intervention components. Perceptions on training appropriateness, effectiveness and methods. |
| Study participants from high-risk cohort in intervention arm | Perception appropriateness of anti-stigma content and methods for adolescents. Perceptions about impact and effectiveness of anti-stigma campaign in community. Any reported change in perception about mental health-related stigma and help seeking in self or family. |
|  | Perception about case detection, treatment and follow-up through EDSS. Facilitators and barriers in help-seeking by adolescents experience of care and perception about quality of care, provided by doctors. Positive/negative experiences as a study participant. Perception about benefits/effectiveness of the intervention. Any spillover effects experience of helping or sharing information with other adolescents not in the study cohort. |
| Study participants from non-high-risk cohort in intervention arm | Perceptions about impact and effectiveness of anti-stigma campaign in community. Any reported changes in perception about mental health-related stigma and help seeking in self or family. Facilitators and barriers in treatment seeking. Positive/negative experiences as a study participant. Perception about benefits/effectiveness of the intervention. Any spillover effects experience of helping or sharing information with other adolescents not in the study cohort. |
| Study participants from (high-risk and non-high-risk cohort) in the control arm | Perception about impact and effectiveness of anti-stigma content. Any reported change in perception about mental health-related stigma and help seeking in self or family. Facilitators and barriers in help seeking. Experience as a study participant. |
| Parents of adolescents in the study cohort | Perception about impact and effectiveness of anti-stigma content. Any reported change in perception about mental health-related stigma and help seeking in self or family. |
|  | Knowledge about key stressors and symptoms of mental health problems in adolescents. Facilitators and barriers in help-seeking by adolescents. Experience of help seeking. Perception about benefits/effectiveness of the intervention. |
| Adolescent expert Advisory Groups in each site | Perception about their role on co-creation of anti-stigma material and feedback on the whether the anti-stigma material was able reflect their suggestions. Perception about impact and effectiveness of anti-stigma content among adolescents and community members. Any reported change in perception about mental health-related stigma and help seeking in self or family. Perceived personal benefit/challenges in being an AEAG member. Perception about benefit and effectiveness of mHealth component. Facilitators and barriers in help seeking by high-risk adolescents. Suggestions on improving engagement/involvement of younger adolescents in AEAGs and the cocreation process. |
| Peer-group intervention arm | Perceptions on effectiveness and appropriateness of training received. Perceptions about impact and effectiveness of anti-stigma campaign in community. Perception about effectiveness anti-stigma campaign in improving help seeking by adolescents. Facilitators and barriers in treatment seeking by high-risk cohort. Experience if any of providing help to adolescents in the community who reached out to the peer group. Overall experiences of participating in the trial. How does the learnings from the project/trainings will be useful beyond the project period. |
| Doctors from intervention arm | Perceptions on effectiveness and appropriateness of training received. Experience of using technology-based decision support system for diagnosis and management of depression and self-harm among adolescents. Perceived effectiveness anti-stigma campaign. Possible facilitators and barriers to scaling up mHealth component. Overall experience of participating in the trial including any challenges. Facilitators and barriers in help seeking and treatment adherence among adolescents. Whether learnings from the project/trainings will be useful beyond the project period. |
| Study participants from high-risk cohort in intervention arm who visited the doctor and were given medical and/or psychological treatment. | Experience of seeking care from primary care doctor and perception about quality of care. Level of comfort/discomfort in sharing their mental distress experiences with doctors. Perceived benefit, if any, as a result of treatment. Barriers and facilitators for seeking treatment from a specialist (in case of referral). Views about help-seeking in future or continuing care if advised by the doctor. Positive/negative experiences as a study participant. Suggestions for the programme in the future. |
| Parents/guardians of study participants from high-risk cohort in intervention arm who visited the doctor | Perception about impact and effectiveness of anti-stigma content. Any reported change in perception about mental health-related stigma and help seeking in self or family. Knowledge about key stressors and symptoms of mental health problems in adolescents. Knowledge perception about mental health problems of their ward. Barriers and facilitators for seeking mental healthcare. |
|  | Experience of seeking mental health care for their wards. Perceived benefit, if any, as a result of treatment. Views about help-seeking in future or continuing mental health care if advised by the doctor. Suggestions for the programme in the future. |

**Table 3** Continued

| Type of group/individual | Some areas of inquiry |
|---|---|

AEAGs, Adolescent Expert Advisory Groups; EDSS, electronic decision support system.

involves an anti-stigma campaign and a mobile device-based decision support system for primary care doctors and NPHWs, to improve treatment of adolescents at high risk of depression, other significant emotional or medically unexplained complaints and suicide risk.

The process evaluation will help us understand and explain key causal mechanisms that led to change and will, therefore, strengthen the understanding of the implementation process by highlighting various barriers and facilitators. Furthermore, it will provide an understanding of how the local context played a role in the way the intervention was implemented and help identify the need and impact of any adaptations made to the intervention. The process evaluation will provide stakeholder perspectives on aspects of the intervention that worked and those that need further adaptation.

## Strengths and limitations

The protocol was developed using existing implementation science theories and frameworks and combines qualitative and quantitative measures to understand key aspects of how the intervention was implemented and which aspects worked or need further improvement. The MRC framework, which is a comprehensive framework, widely used in the field of implementation science will be used. The case study method using a maximum variation approach will be used to make suitable comparisons among different contexts and avoid Hawthorne. The ARTEMIS intervention does not directly address social determinants which impact mental health outcomes. In this study, we will collect additional qualitative and quantitative data on social support, socioeconomic conditions, education and the local context of participants which will help to better understand and explain linkages between mental health outcomes and social determinants.

## Ethics and dissemination

The study received formal ethics approval from the Ethics Committee of the George Institute for Global Health India on 4 September 2020 (project number 17/2020). The study also received formal ethics approval from the Research Governance and Integrity Team, Imperial College, London on 8 June 2022 (ICREC reference number: 22IC7718). The Health Ministry's Screening Committee and Indian Council for Medical Research have also provided approval to the project (ID 2020-9770). Findings will be disseminated to study participants and other stakeholders at a policy symposium. All identifiable personal data will be stored in password-protected secured servers located at The George Institute for Global, India office in Hyderabad. Only deidentified data will be disseminated. Data will be available with the Principal Investigator (PI) on an accessible data repository, which can be accessed by other researchers, subject to a formal request to the PI to access the data for research purposes.

## Trial status

At the time of writing the paper, the intervention had started in both sites. Randomisation was executed in Vijayawada and in Delhi on 12 December 2022. Intervention components were implemented in both the sites.

**Author affiliations**
[1]The George Institute for Global Health India, New Delhi, India
[2]The George Institute for Global Health, Hyderabad, India
[3]Inflammation Biology, King's College London, London, UK
[4]Department of Communication, University of Hyderabad, Hyderabad, India
[5]Dr.A.V. Baliga Memorial Trust, New Delhi, India
[6]Institute of Health Policy, Management and Evaluation, University of Toronto, Toronto, Ontario, Canada
[7]Department of Psychiatry, All India Institute of Medical Sciences, New Delhi, India
[8]Young Lives, India, New Delhi, India
[9]The George Institute for Global Health, Sydney, New South Wales, Australia
[10]University of New South Wales, Sydney, New South Wales, Australia
[11]Imperial College London, London, UK
[12]Centre for Global Mental Health and Centre for Implementation Science, Institute of Psychiatry, Psychology and Neuroscience, King's College London, London, UK

**Acknowledgements** We acknowledge the contribution of all members of the Adolescent Expert Advisory Group in Delhi and Vijayawada who have generously contributed their time and expertise to the development of ARTEMIS. GT is supported by the National Institute for Health and Care Research (NIHR) Applied Research Collaboration South London (NIHR ARC South London) at King's College Hospital NHS Foundation Trust.

**Contributors** The paper was conceptualised by PKM. AM wrote the original draft with contributions from SKY. PKM commented on multiple drafts before sending a prefinal version to everyone listed as authors. SK, SP, HL, UR, BME, RaS, ReS, DP, RN and GT reviewed the draft and provided critical intellectual inputs and comments to the draft. PKM led the implementation of the trial in India along with SKY, SK, SP and AM. SKY, SK and SP played a key role in implementing research activities in the sites. All authors read and approved the final manuscript.

**Funding** Adolescents' Resilience and Treatment nEeds for Mental health in Indian Slums (ARTEMIS) is funded by UK Research and Innovation/Medical Research Council (UKRI/MRC), Grant no: MR/S023224/1. SKY, SK and SP are all partially or fully supported by ARTEMIS. PKM and AM are partially supported through NHMRC/GACD grant (SMART MentalHealth-APP1143911) and ARTEMIS. PKM is the PI for ARTEMIS and Co-PI for SMART Mental Health. HL and GT are partly support by ARTEMIS. GT is also supported by the National Institute for Health Research (NIHR) Applied Research Collaboration South London at King's College London. GT is also supported by the MRC-UKRI in relation to the Indigo Partnership (MR/R023697/1) awards.

**Disclaimer** The views expressed are those of the author(s) and not necessarily those of the NHS, the NIHR or the Department of Health and Social Care. The funding agency had no role in the writing of the manuscript or the decision to submit it for publication.

**Competing interests** We declare that the George Institute for Global Health has a part-owned social enterprise, George Health Enterprises, which has commercial relationships involving digital health innovations.

**Patient and public involvement** Patients and/or the public were involved in the design, or conduct, or reporting, or dissemination plans of this research. Refer to the Methods section for further details.

**Patient consent for publication** Not applicable.

**Provenance and peer review** Not commissioned; externally peer reviewed.

**Open access** This is an open access article distributed in accordance with the Creative Commons Attribution 4.0 Unported (CC BY 4.0) license, which permits others to copy, redistribute, remix, transform and build upon this work for any purpose, provided the original work is properly cited, a link to the licence is given, and indication of whether changes were made. See: https://creativecommons.org/licenses/by/4.0/.

**ORCID iDs**
Ankita Mukherjee http://orcid.org/0000-0002-6236-1317
Sandhya Kanaka Yatirajula http://orcid.org/0000-0002-0329-7271
Srilatha Paslawar http://orcid.org/0000-0002-9005-2810
David Peiris http://orcid.org/0000-0002-6898-3870
Pallab Kumar Maulik http://orcid.org/0000-0001-6835-6175

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
