## [Reviewer comments · BMJ Open]

ARTICLE DETAILS

TITLE (PROVISIONAL)	Protocol for process evaluation of ARTEMIS cluster randomized controlled trial: an intervention for management of depression and suicide among adolescents living in slums in India
AUTHORS	Mukherjee, Ankita; Yatirajula, Sandhya Kanaka; Kallakuri, Sudha; Paslawar, Srilatha; Lempp, Heidi; Raman, Usha; Essue, Beverley; Sagar, Rajesh; Singh, Renu; Peiris, David; Norton, Robyn; Thornicroft, Graham; Maulik, Pallab

VERSION 1 – REVIEW

REVIEWER	Worrell, Courtney King's College London, Psychological Medicine
REVIEW RETURNED	18-Jan-2024

GENERAL COMMENTS	This paper presents a very interesting protocol for process evaluation, and I was delighted to review it. Overall, the paper is very well written, and I only have some minor suggestions which the authors may consider to help make sure all of the relevant information is as clear as possible throughout. 1. While the abstract presents a good summary of the protocol paper, I might suggest a couple of minor edits which I feel will help to emphasise some key points from the protocol. Firstly, in the 'Introduction' section of the Abstract, the authors introduce that 'The aim of this paper is to describe the process evaluation protocol for the ARTEMIS trial' which I would suggest developing a little further to provide an explanation for the purpose of the process evaluation is being performed given that this is the central focus of the protocol. Similarly, I think the 'Methods' section of the Abstract would benefit from a little more information on the analysis which will be conducted in addition to the guidance which is currently presented.2. In the 'Data Collection' section of the Methods, the authors might consider expanding on what is meant by 'Data on competency and fidelity measures will also be captured throughout the intervention.' (page 10, lines 35-36). It would be helpful for readers to understand what measures will be used. If this information refers to the section on 'Assessment of Implementation outcomes' (page 11) then for clarity, it might be helpful to refer to this.3. Also in the 'Data Collection' section of the 'Methods,' it may be useful to state how many participants will be in each focus group and whether there is any sample size justification for the qualitative data collection (paragraph starting on line 39, page 10).
--

	4. In the 'Data Analysis' section of the methods, I might also suggest that the authors could expand on what analysis will be performed with the quantitative data. The approach that will be taken to this analysis is not entirely clear to me. 5. I would also suggest that the authors are consistent about the strengths and limitations of study as some points mentioned in the bullet points on page 4 are not presented in the later section. Thank you for the opportunity to review the paper.
--	--

REVIEWER	Armstrong, Gregory University of Melbourne, Centre for Mental Health, Melbourne School of Population and Global Health
REVIEW RETURNED	08-Mar-2024

GENERAL COMMENTS	Thank you for the opportunity to review this protocol paper, focused on an implementation science evaluation of a slum-based intervention to reduce mental health stigma, improve help-seeking and improve mental health-related clinical decision making among primary care doctors. I found it to be well-written and a strong description of the methods that are being employed by the investigators. Some very minor comments/edits:  1. My main remark, which the authors obviously can't do anything about at this stage, is that the intervention is focused on mental health in quite a narrow sense. In their vision it is about reducing stigma and improving access to clinical care. These are extremely worthwhile goals, of course, and I should stop there. I just couldn't help but comment on the social determinants aspects of mental health for this slum-based community, and wondered how these might be understood within the context of this intervention. As per Kleinman's theory of social suffering, we can sometimes get into the situation where we're pathologizing individuals for what are quite "normal" responses to social circumstances. It would be nice to see a sentence or two in the introduction that makes further acknowledge of social determinants of mental health. I think it's in part why several of the other community-based interventions for youth in slums have focused on things like building resilience or life skills, as it is an acknowledgement that these things are especially important when in living in such precarious social/economic/environmental etc circumstances. 2. Page 5, Line 11, an 's' seems to be needed after the word adolescent. 3. Page 5, lines 53-60. The definition of a slum should probably go in the methods section 4. Page5-6 describes 'ARTEMIS as a community-based cluster randomised control trial that aims to reduce depression and risk of suicide among adolescents living in slums'. After reading that, I was confused reading the methods, and I was at first looking for what the primary outcome variable was and other elements you might expect in a trial of intervention effectiveness. However, the methods and study aims (page 6, lines 25-40) indicate implementation science. I might just ask the authors to consider the wording in the sentence I mention above as it may cause some minor confusion for readers. It might be about clearly drawing out the aim of the intervention, versus the aim of the study (I realise
---

	the authors have done this elsewhere in the manuscript, including the title). 5. Page 7, line 38: it seems the word “based” is redundant and should be removed 6. Page 10, line 25: a full stop is needed after the word “information”.
--	--

VERSION 1 – AUTHOR RESPONSE

Reviewer 1. Comments

i) in the ‘Introduction’ section of the Abstract, the authors introduce that ‘The aim of this paper is to describe the process evaluation protocol for the ARTEMIS trial’ which I would suggest developing a little further to provide an explanation for the purpose of the process evaluation is being performed given that this is the central focus of the protocol.

Response: Added a para to explain the role of objective of process evaluation.

ii) the ‘Methods’ section of the Abstract would benefit from a little more information on the analysis which will be conducted in addition to the guidance which is currently presented.

Response: Explained how data analysis will be done

iii) In the ‘Data Collection’ section of the Methods, the authors might consider expanding on what is meant by ‘Data on competency and fidelity measures will also be captured throughout the intervention.’ (page 10, lines 35-36). It would be helpful for readers to understand what measures will be used. If this information refers to the section on ‘Assessment of Implementation outcomes’ (page 11) then for clarity, it might be helpful to refer to this.

Response: The lines on pg 10 refer to the information on pg 11. To make things clearer, and make the language consistent, the section title ‘Assessment of Implementation Outcomes’ has been changed to ‘Assessment of Implementation Fidelity’

iv) in the ‘Data Collection’ section of the ‘Methods,’ it may be useful to state how many participants will be in each focus group and whether there is any sample size justification for the qualitative data collection (paragraph starting on line 39, page 10). It is difficult to give exact planned numbers for each FGD.

Response: In the text we have added that each FGD will have 6-12 participants (pg 10). We do not find it useful to mention the exact number of participants under each category at this stage. Literature on FGDs as a method recommend against having more than 12 participants for a meaningful discussion. To the best of our knowledge, there is no thumb rule for a proportion of participants who should be included in one FGD since we are using nonprobability sampling.

v) In the ‘Data Analysis’ section of the methods, I might also suggest that the authors could expand on what analysis will be performed with the quantitative data. The approach that will be taken to this analysis is not entirely clear to me.

Response: The quantitative data will include data from the open MRS. This will primarily include data on key processes in the programme like visits made by non-physician health workers number of visits to doctor, referral data and so on. The data analysis will only use descriptive statistics for the process evaluation. More details are provided under the trial protocol (Ref #22)

vi) I would also suggest that the authors are consistent about the strengths and limitations of study as some points mentioned in the bullet points on page 4 are not presented in the later section.

Response: The points on page 4 have been added under ‘Discussions- Strengths and limitations’ (Pg 12)

Reviewer 2 Comments

i) the intervention is focused on mental health in quite a narrow sense. In their vision it is about reducing stigma and improving access to clinical care. These are extremely worthwhile goals, of

course, and I should stop there. I just couldn't help but comment on the social determinants aspects of mental health for this slum-based community, and wondered how these might be understood within the context of this intervention. As per Kleinman's theory of social suffering, we can sometimes get into the situation where we're pathologizing individuals for what are quite "normal" responses to social circumstances. It would be nice to see a sentence or two in the introduction that makes further acknowledge of social determinants of mental health. I think it's in part why several of the other community-based interventions for youth in slums have focused on things like building resilience or life skills, as it is an acknowledgement that these things are especially important when in living in such precarious social/economic/environmental etc circumstances. We agree that social determinants play a critical role in the mental health of adolescents living in slums.

Response: The focus of ARTEMIS was on testing strategies (mHealth) to strengthen mental service delivery, and address stigma which is a key demand side barrier in accessing these services. While the intervention and main outcomes do not specifically focus on addressing social determinants affecting the mental health outcomes, we will collect additional qualitative and quantitative data on social support, socioeconomic conditions, education, and the local context of participants which will help to draw linkages between mental health and social determinants. This would be discussed in future outputs from this project.

We have added this as one of the limitations in paper(pg13).

ii) Page 5, Line 11, an 's' seems to be needed after the word adolescent.

Response: Added 's'

iii) Page 5, lines 53-60. The definition of a slum should probably go in the methods section

Response: Moved definition slum and ward from introduction (pg5) to Methods- Study setting (pg7)

iv) Page5-6 describes 'ARTEMIS as a community-based cluster randomised control trial that aims to reduce depression and risk of suicide among adolescents living in slums'. After reading that, I was confused reading the methods, and I was at first looking for what the primary outcome variable was and other elements you might expect in a trial of intervention effectiveness. However, the methods and study aims (page 6, lines 25-40) indicate implementation science. I might just ask the authors to consider the wording in the sentence I mention above as it may cause some minor confusion for readers. It might be about clearly drawing out the aim of the intervention, versus the aim of the study (I realise the authors have done this elsewhere in the manuscript, including the title).

Response: We have first added a line to say that this paper discusses the protocol for process evaluation of the ARTEMIS cRCT. This is followed by the description of the intervention. This should help readers understand that we are describing the intervention for which process evaluation will be done. Hopefully that will help lay out the context of this paper and the aims as relevant to this paper. The main trial protocol paper has been published earlier (Ref #22) This has also been mentioned in numerous other places including abstract and title as noted by the reviewer.

v) Page 7, line 38: it seems the word "based" is redundant and should be removed

Response: 'based' deleted

vi) Page 10, line 25: a full stop is needed after the word "information".

Response:Full stop added

We would like to bring to your notice that we have made a minor revision to 'Strengths and Limitations' and the abstract. This has been provided as tracked change in the main document marked copy.

With Warm Regards
Prof. Pallab K Maulik
(Corresponding Author)

VERSION 2 – REVIEW

REVIEWER	Worrell, Courtney King's College London, Psychological Medicine
REVIEW RETURNED	26-Mar-2024

GENERAL COMMENTS	I would like to thank the authors for taking the time to address my reviewer comments for the paper 'Protocol for process evaluation of ARTEMIS cluster randomized controlled trial: an intervention for management of depression and suicide among adolescents living in slums in India'. I am happy that most of my comments have been addressed. I have one remaining point regarding my 4th point made regarding the 'Data Analysis' section of the 'Methods' and how it would be beneficial for the authors to expand on the analysis that will be performed in the quantitative data. I appreciate the response to the comment explaining the use of the data however in my opinion, I still think there would be benefit from clarifying this in this section of the methods, even if to briefly emphasise that more details are provided in the reference given. Thank you again for the opportunity to review this paper.
---

REVIEWER	Armstrong, Gregory University of Melbourne, Centre for Mental Health, Melbourne School of Population and Global Health
REVIEW RETURNED	25-Mar-2024

GENERAL COMMENTS	The authors have nicely addressed all my concerns/comments.
---

VERSION 2 – AUTHOR RESPONSE

Reviewer Comment	Response
I am happy that most of my comments have been addressed. I have one remaining point regarding my 4th point made regarding the 'Data Analysis' section of the 'Methods' and how it would be beneficial for the authors to expand on the analysis that will be performed in the quantitative data. I appreciate the response to the comment explaining the use of the data however in my opinion, I still think there would be benefit from clarifying this in this section of the methods, even if to briefly emphasise that more details are provided in the reference given.	The section on analysis has been restructured and an effort has been made to clarify the frameworks and approach to the quantitative analysis. Table 2 which provides the overall conceptual framework, indicates the key areas of enquiry as well as data sources can provide further clarity. For example, one area of inquiry is to understand 'Reach' of the intervention. Therefore 'number and proportion of high-risk cohort reached by the intervention' will be one of the several indicators we will study. Similarly, to understand 'Acceptability/Adoption' of intervention we will look at data on 'average time taken for diagnosis using the app'. All this data is available in OpenMRS and was collected during the intervention. Due to

	the nature of this data which is process data, data analysis will only use descriptive statistics like proportions, average, range for these.
--	---